# Genomic Selection for Live Weight in the 14th Month in Alpine Merino Sheep Combining GWAS Information

**DOI:** 10.3390/ani13223516

**Published:** 2023-11-14

**Authors:** Chenglan Li, Jianye Li, Haifeng Wang, Rui Zhang, Xuejiao An, Chao Yuan, Tingting Guo, Yaojing Yue

**Affiliations:** 1Key Laboratory of Animal Genetics and Breeding on Tibetan Plateau, Ministry of Agriculture and Rural Affairs, Lanzhou Institute of Husbandry and Pharmaceutical Sciences, Chinese Academy of Agricultural Sciences, Lanzhou 730050, China; 82101215420@caas.cn (C.L.);; 2Sheep Breeding Engineering Technology Research Center of Chinese Academy of Agricultural Sciences, Lanzhou 730050, China

**Keywords:** genomic selection, GWAS, prior marker information, prediction accuracy

## Abstract

**Simple Summary:**

The weight features of Alpine Merino Sheep, particularly their live weight in the 14th month (WT), are crucial in production and breeding, as they live at high altitudes and in highland areas where oxygen is scarce. Genome selection (GS) offers the advantages of minimizing breeding costs and generational intervals, and the main objective of genome selection breeding is the accuracy of GS. Genome-wide association studies (GWASs) were used in this study to extract the variation information related to the 14-month-old body weight trait of Alpine Merino Sheep, identify candidate genes associated with this trait, and employ GWAS results as prior information to conduct GS research on the trait to determine the most suitable GS breeding model for the target trait.

**Abstract:**

Alpine Merino Sheep is a novel breed reared from Australian Merino Sheep as the father and Gansu Alpine Fine-Wool Sheep as the mother, living all year in cold and arid alpine areas with exceptional wool quality and meat performance. Body weight is an important economic trait of the Alpine Merino Sheep, but there is limited research on identifying the genes associated with live weight in the 14th month for improving the accuracy of the genomic prediction of this trait. Therefore, this study’s sample comprised 1310 Alpine Merino Sheep ewes, and the Fine Wool Sheep 50K Panel was used for genome-wide association study (GWAS) analysis to identify candidate genes. Moreover, the trial population (1310 ewes) in this study was randomly divided into two groups. One group was used as the population for GWAS analysis and screened for the most significant top 5%, top 10%, top 15%, and top 20% SNPs to obtain prior marker information. The other group was used to estimate the genetic parameters based on the weight assigned by heritability combined with different prior marker information. The aim of this study was to compare the accuracy of genomic breeding value estimation when combined with prior marker information from GWAS analysis with the optimal linear unbiased prediction method for genome selection (GBLUP) for the breeding value of target traits. Finally, the accuracy was evaluated using the five-fold cross-validation method. This research provides theoretical and technical support to improve the accuracy of sheep genome selection and better guide breeding. The results demonstrated that eight candidate genes were associated with GWAS analysis, and the gene function query and literature search results suggested that *FAM184B*, *NCAPG*, *MACF1*, *ANKRD44*, *DCAF16*, *FUK*, *LCORL*, and *SYN3* were candidate genes affecting live weight in the 14th month (WT), which regulated the growth of muscle and bone in sheep. In genome selection analysis, the heritability of GBLUP to calculate the WT was 0.335–0.374, the accuracy after five-fold cross-verification was 0.154–0.190, and after assigning different weights to the top 5%, top 10%, top 15%, and top 20% of the GWAS results in accordance with previous information to construct the G matrix, the accuracy of the WT in the GBLUP model was improved by 2.59–7.79%.

## 1. Introduction

Genomic selection (GS), a novel method for selective breeding using high-density markers covering the whole genome, was initially proposed by Meuwissen in 2001 [1]. Particularly for traits with low heritability and difficulty in measurement, GS offers the advantages of shortening the generation interval, achieving early selection, reducing breeding cost [2,3,4], and improving the estimated accuracy of the genomic estimated breeding value (GEBV) [5]. It partially resolves the issues with marker-assisted selection and completes the task of using genome technology to direct breeding practice. The direct method represented by genomic best linear unbiased prediction (GBLUP) and the indirect method represented by the Bayesian method are the two primary GS methods [1]. Despite having the advantage of high calculation accuracy, the indirect method’s application in production is hampered by the issues of having a long calculation time and large consumption of computing resources; meanwhile, GBLUP offers the benefits of rapid computation and user-friendly implementation. It has been employed in the breeding of pigs [6,7], cattle [8], sheep [9], chickens [10], and other poultry. Recently, GS has been extensively used to study the quantitative traits of livestock, which has been attributed to the decreasing cost of high-throughput sequencing. Studies on important economic traits of livestock and poultry report that integrating previous data of GWAS can improve GS accuracy to some extent [11].

Genome-wide association analysis (GWAS) is a method of identifying potential variants associated with target traits and the candidate genes associated with them, and it provides evidence-based breeding production theories. N. Risch and K. Merikangas first proposed GWAS while researching the genetics of complicated diseases. Various software packages and models are available to conduct GWAS, including the most commonly used general linear model (GLM), mixed linear model (MLM), FarmCPU model, etc. Each model has its advantages and disadvantages. In this study, the FarmCPU model in the rMVP package (https://github.com/xiaolei-lab/rMVP (accessed on 15 May 2023)) was used for analysis, which combines the advantages of the MLM and GLM and, through iteration, uses them to overcome their shortcomings. Eliminating the confusion between relatives and target trait genes in MLMs can reduce errors more effectively.

Alpine Merino Sheep is a novel breed that was successfully developed with modern breeding technology using Australian Merino Sheep as the father and Gansu Alpine Fine Wool Sheep as the mother. The successful development of this breed not only improved the structure of China’s native sheep breed resources but also significantly boosted the local economy in the cold, arid mountain regions. Body weight is an important growth and development indicator in sheep breeding, which, to some extent, can affect the production of wool and meat [12]. Currently, GS and GWAS are relatively mature for guiding animal breeding. Some researchers have used the GWAS method to identify a series of candidate genes related to the growth, weight, and body size of sheep [13,14,15,16], providing theoretical and methodological support for related work on sheep breeding. Previous studies on human complex traits [17], important economic traits in livestock [18,19,20], and simulation studies [18] have shown that integrating prior GWAS information can improve the accuracy of genomic selection (GS). However, despite several methods based on GS accuracy for important economic traits in sheep existing, only a few studies have used the results of GWAS analysis as prior information in GS to predict accuracy. Meanwhile, given some differences in weight traits between Alpine Merino Sheep and Australian Merino Sheep, this study used the body weight trait of Alpine Merino Sheep at 14 months of age as the target trait and performed GWAS analysis on it, screened out candidate genes related to it, and sorted them by *p*-value significance according to the GWAS analysis results. It further selected the results of top 5%, top 10%, top 15%, and top 20% SNPs as prior information to construct a G matrix in GS to evaluate the accuracy of breeding values to identify candidate genes associated with the 14-month-old weight trait of Alpine Merino Sheep and compare the accuracy of genomic breeding value estimation combined with the prior GWAS marker information and method for the breeding value estimation of 14-month-old merino sheep. This research can to some extent accelerate the selection and breeding of Alpine Merino Sheep while also providing theoretical support for subsequent sheep breeding.

## 2. Materials and Methods

### 2.1. Phenotype and Genotype Data

The experimental animals in this study were chosen from the Gansu Sheep Breeding Technology Promotion Station in Huangcheng Town, Sunan Yugu Autonomous County, Zhangye City, Gansu Province. The station maintains a strictly standardized breeding and management system to ensure that all animals receive consistent feeding and care. All the experimental animals in this study were born in the same year and month, and they have the same gender but come from different flocks. The age of each participant was restricted to 14 months to ensure the uniformity of the sample. To prevent inaccuracies, after 12 h of fasting in their natural conditions, we measured the weight of each sheep by placing them on an automated weighing scale. The measurement unit is kilograms. The results were rounded to one decimal place for precision, and outliers (extreme values) were excluded from the analysis. The outcomes were systematically archived according to the documentation by proficient and experienced technicians. Outliers and extreme values were removed using the three standard deviations method, and all 1310 ewes had complete records of their phenotypic data. The entire population followed a the normal distribution. In this study, the weight phenotype was divided into two parts: (1) to identify candidate genes associated with live weight at 14 months, a total of 1310 Alpine Merino ewes were used for GWAS analysis. (2) To compare the genomic prediction accuracy between the GBLUP model and the GBLUP model incorporating GWAS information, the phenotypes of 1310 ewes were randomly divided into two groups, each containing 655 ewes, using the RAND function. These two groups were considered control groups for each other.

Blood samples (of at least 3 mL) were collected from each sheep by puncturing the jugular vein. The sample were then immediately transferred to vacuum sampling vessel with ethylenediaminetetraacetic acid (EDTA) added (Yuli Medical Equipment Co., Ltd., Taizhou, China). Afterward, blood samples were stored in the onboard refrigerator at 4 °C for subsequent DNA extraction. Genotyping was conducted using the Fine Wool Sheep 50K Panel (MolBreeding Biotech Ltd., Shijiazhuang, China). The reference genome used was the Oar v4.0 (GCF 000298735.2) version of sheep, as published on the NCBI. The PLINK (version: v1.9b4) conducted quality control on all SNPs. For SNP sites, the deletion rate was >10%, the test *p*-value (HWE) of the Hardy–Weinberg balance was <10^−6^, and all the markers with a minor allele frequency (MAF) of <5% were eliminated. For sample individuals, all samples with an undetected rate > 10% were eliminated. To sum up, this study’s quality control criteria were as follows: SNP Call rate > 90%, MAF > 5%, HWE_*p*-value > 10^−6^, and Individual Call Rate > 90%. Beagle software (version 5.0) performs the self-filling of genotype data, since some models do not allow the presence of NA in genotype data. A total of 41,987 autosomal SNPs were isolated from the Fine Wool Sheep 50 K, and all 1310 samples passed the quality control criteria; then, they were retained for further GWAS and GS analysis following quality control and genotype data filling.

### 2.2. GWAS Model Analysis Methods

This study utilized the FarmCPU model in rMVP software (version 1.0.7) (https://github.com/xiaolei-lab/rMVP, accessed on 15 May 2023) to perform GWAS analysis for live weight at 14 months. To eliminate the ambiguity of determining whether a relevant marker is in linkage disequilibrium (LD) with a detection marker, the FarmCPU model addresses kinship confusion by adopting a fixed-effect model. The approach removes the need to collect kinship information from all markers or related labels. Alternatively, the related marker is selected using the maximum likelihood approach based on kinship derived from the related marker. This method overfits the stepwise regression model. The FarmCPU model iterations utilize both a fixed-effect model and a random-effects model. In summary, the FarmCPU model is not only a statistically powerful method but also a computationally efficient GWAS approach [21]. The model is represented as follows:(1)$$y_{i}=M_{i1}b_{1}+M_{i2}b_{2}+\ldots +M_{in}b_{n}+Z_{ij}u_{j}+e_{i}$$
(2)$$y_{i}=V_{i}+e_{i}$$

In Equations (1) and (2), Equation (1) is a fixed-effect model, Equation (2) is a random-effects model, $y_{i}$ represents the phenotypic observation of the ith individual, $M_{in}$ represents the classification result of a total of n potential correlation sites included in the model, $b_{n}$ indicates the effect value corresponding to the site, $Z_{ij}$ indicates the classification result of the jth marker of the ith individual, $u_{j}$ is the effect value of $Z_{ij}$, $V_{i}$ represents the total genetic effect of the ith individual, and $e_{i}$ is the residual vector, subject to $e\sim N\left(0,I\sigma_{e}^{2}\right)$. When executing the FarmCPU model, styles (1) and (2) are alternate operations.

The principal component analysis (PCA) results were evaluated using the BLINK model in the GAPIT software (version 3), and the FarmCPU model integrates both kinship relationships and population structure. The results of the GWAS analysis were corrected using the Bonferroni [22] correction method, which strictly controls the occurrence of false positives. The significance threshold for the genome was set at 1/Nsnp, where Nsnp represents the number of SNPs remaining after quality control. The gene annotation used in this study was based on the reference genome of sheep Oar v4.0 (GCF 000298735.2) published on the National Center for Biotechnology Information Website, and the gene annotation software was the BEDTOOLS software (version 2.30.0). The *p*-value of each SNP was obtained using a significance test, utilizing each SNP as a fixed factor for regression analysis. SNPs with smaller *p*-values were then selected and weighted according to their heritability as prior marker information for the subsequent GS analysis.

### 2.3. GS Model Analysis Method

In this study, the GBLUP [23] model in ASReml [24] software (version 4.1.0.176) was used for analysis, and the GBLUP model is a representative of the direct method in GS analysis, which takes individuals as random effects, constructs a kinship matrix as a variance–covariance matrix concerning the genetic information of the population and the predicted population, estimates the variance component by an iterative method, and further solves the mixed model to obtain the estimated breeding value of the individual to be predicted. In this study, the genome was included as a random factor, while birth type and flock were included as fixed factors in the model for analysis. The GBLUP model is as follows:(3)y=Xb+Zu+e

In Equation (3), y represents the corrected tabular vector of the individual, b represents the fixed effect vector (including the overall mean), u represents the random effects vector, i.e., the random additive genetic effect, X and Z represent the association matrix corresponding to b and u, respectively, e represents the random residual, and its variance distribution is $e\sim N\left(0,I\sigma_{e}^{2}\right)$, where lie the identity matrix and the covariance matrix of the additive effects $Var\left(u\right)=G\sigma_{e}^{2}$. Thus, it is necessary to construct a matrix, which is the genomic relationship matrix between individuals, based on the method proposed by VanRaden [23], which is as follows:(4)$$G=\frac{\sigma \sigma^{T}}{2\sum_{k=1}^{m}pk\left(1-pk\right)}\;\;$$

In Equation (4), σ is the additive genetic effect marker matrix, and its dimension is the sum of the number of individuals (n) and the number of the loci (m), $\sigma^{T}$ is the transpose matrix of σ, and $p_{k}$ is the minor allele frequency (MAF) value of the gene seat (k).

The kinship matrix construction method used by GS combined with GWAS prior information is as follows. The kinship matrix $G_{1}$ was constructed using the significance prior marker information in the GWAS results, and the kinship matrix $G_{2}$ was constructed from the remaining SNP sites, the genetic variance of $G_{1}$ and $G_{2}$ for WT trait interpretation was calculated individually, and the kinship matrix $G_{3}$ was fitted with the proportion of explanatory genetic variance as the weight. The $G_{3}$ matrix equation is as follows:(5)$$G_{3}=\gamma G_{1}+\left(1-\gamma \right)G_{2}$$

In Equation (5), $\gamma =\frac{\sigma_{G1}^{2}}{\sigma_{G1}^{2}+\sigma_{G2}^{2}}$, $\sigma_{G1}^{2}$ is the genetic variance of the $G_{1}$ matrix, and $\sigma_{G2}^{2}$ is the genetic variance of the $G_{2}$ matrix. GS analysis was performed using kinship matrix G and kinship matrix $G_{3}$ combining different prior marker information, and genetic parameters and the prediction accuracy of the WT trait were compared between different methods.

### 2.4. Calculation Method of GS Accuracy

The accuracy of genomic predictions was assessed using five-fold cross-validation [25]. The 1310 datasets were randomly divided into two groups, each containing 655 ewes. Five-fold cross-validation was performed, with 655 ewes were divided into five subsets of approximately equal numbers, with each subset containing 131 individuals. Four of these groups were considered as training sets (reference populations) for each five-fold cross-validation to evaluate genetic parameters and genomic breeding values. The remaining subset was considered as a validation population (candidate population) to ensure accuracy and predict the breeding value of the validation population. To ensure the randomness of the validation population, the cross-validation mentioned above was conducted ten times. Subsequently, the average of the five-fold cross-validations was calculated and recorded as the final prediction accuracy. Currently, in genomic prediction selection, accuracy is mainly obtained by calculating the Pearson correlation coefficient between GEBV and TBV, the computational method was proposed by Meuwissen in 2001 [1],and the accuracy is calculated as follows:(6)$$R=\frac{\mathrm{Cov}\left(GEBV,TBV\right)}{\sqrt{Var\left(GEBV\right)\times Var\left(TBV\right)}}$$

In Equation (6), $\mathrm{Cov}\left(GEBV,TBV\right)$ represents the covariance of GEBV and TBV, whereas $Var\left(GEBV\right)$ and $Var\left(TBV\right)$ represent the variance of GEBV and TBV, respectively. Since this calculation applies to populations whose true values are known, our study is entirely relevant.

## 3. Results

### 3.1. Statistics and Distribution of Phenotypic Data

In this study, the weight traits of 1310 Alpine Merino Sheep ewes were recorded at 14 months of age by professional and experienced staff. As indicated by the pink line, most of the records were concentrated between 40 and 50 kg, as depicted in Appendix A. Live weight in the 14th month was assessed using the debug function in the psych package of the R language. The descriptive statistical results are shown in Table 1 (includes overall phenotype and grouped phenotype).

### 3.2. Call Rate and Distribution of Genotype Data

After quality control and genotype filling of the Fine Wool Sheep 50K data, a total of 41,987 SNPs were utilized for genome-wide association analysis and genomic selection. The *p*-values from the GWAS analysis were sorted in descending order, and the top 5%, top 10%, top 15%, and top 20% of SNPs were selected as prior marker information. The top 5% contained 1908 SNPs, leaving 36,568 SNPs in the remaining 95%. The top 10% contained 3835 SNPs, leaving 34,642 SNPs in the remaining 90%. The top 15% contained 5726 SNPs, leaving 32,750 SNPs in the remaining 85%. The top 20% contained 7666 SNPs, leaving 30,810 SNPs in the remaining 80%. All of these SNPs were distributed across each chromosome.

All the trial populations used in this study came from five different core breeding farms. After quality control, a total of 41,987 autosomal SNPs were used for principal component analysis (PCA), The PCA diagram is exhibited in Figure 1a, indicating stratification in the experimental population. PCA was included in the GWAS model for correction therefore. Linkage disequilibrium analysis was conducted on Alpine Merino Sheep, and the results are shown in Figure 1b. When the distance reached 100 kb, LD began to exhibit a trend toward stability, suggesting that genes within 100 kb of the significant SNPs could be potential candidates. The 100 kb region upstream and downstream of significant SNPs was selected to annotate candidate genes in this study.

### 3.3. GWAS Analysis Results

The FarmCPU model was used to perform correlation analysis of the 14-month-old weight trait. The results of the Manhattan plot and QQ plot are shown in Figure 2, the two threshold lines of the Manhattan plot were set to 0.05 (0.05/Nsnp) and 1 (1/Nsnp), and the *p*-value in the GWAS result was corrected by the Bonferroni method. A total of 17 significant sites were associated with a significance threshold of 1, and the upstream and downstream 100 kb intervals of those associated significant loci were extracted to annotate the genes. The annotation tool used was the BEDTOOLS software, which offers a powerful genomic algorithm toolset satisfying the study’s gene annotation requirements, and the results of this study were annotated to a total of 22 adjacent genes (Table 2). In contrast, the SNP loci associated with weight at 14 months were located on chromosomes 1,2, 3, 6, and 14. By querying gene function through online databases and extensively reviewing the literature, a total of eight candidate genes related to live weight in the 14th month were identified. The candidate genes were *ANKRD44* (ankyrin repeat domain 44), *MACF1* (microtubule actin crosslinking factor 1), *SYN3* (synapsin III), *DCAF16* (DDB1 and CUL4-associated factor 16), *FAM184B* (family with sequence similarity 184 member B), *NCAPG* (non-SMC condensing I complex subunit G), *LCORL* (ligand-dependent nuclear receptor corepressor like), and *FUK* (fucose kinase).

### 3.4. GS Analysis Results

The GBLUP model is a representative of the direct method in GS prediction methods, which assumes that all SNPS have an effect. However, this assumption is obviously unreasonable. In this study, group 1 and group 2 were combined to verify the population. When the GBLUP model calculation was not based on the GWAS results, the genetic variance and environmental variance calculated by the G matrix were 5.427~6.176, 10.351 and 10.751, respectively. The heritability was estimated to be between 0.335 and 0.774, and the prediction accuracy, as assessed by five-fold cross-validation, ranged from 0.154 to 0.190. The genetic parameters for group 1 are presented in Table 3, while those for group 2 are shown in Table 4.

The GWAS results from trial group 2 were utilized as prior information to construct a G matrix, which was then used to validate trial group 1 and compare the accuracy of genome prediction. The most significant SNPs (top 5%) were selected to construct the kinship matrix G1. The genetic variance and environmental variance obtained were 2.702 and 13.308, respectively. The heritability was 0.169. The remaining 95% of SNPs were used to construct matrix G2. The genetic variance was 5.418 and environmental variance was 10.764. The heritability was 0.335, and the weighted values of G1 and G2 were 0.333 and 0.667, respectively. The G3 matrix was constructed with genetic variance and environmental variance of 4.840 and 11.229, respectively. The heritability was 0.158, and the prediction accuracy was 0.630, representing a 2.59% improvement in accuracy. The most significant SNPs (top 10%) were selected to construct the kinship matrix G1. The genetic variance and environmental variance obtained were 4.035 and 12.013, respectively. The heritability was 0.251. The remaining 90% of SNPs were used to construct the matrix G2. The genetic variance was 5.256 and the environmental variance was 10.915, resulting in a heritability of 0.325. The weighted values of G1 and G2 were 0.434 and 0.566, respectively. Additionally, the genetic variance was 5.331, the environmental variance was 10.767, the heritability was 0.331, and the prediction accuracy was 0.165, representing a 7.14% improvement in accuracy.

The most significant SNPs (top 15%) were selected to construct the affinity matrix G1. The genetic variance and environmental variance obtained were 4.052 and 11.992, respectively. The heritability was 0.253, and the remaining 85% of the SNPs were used to construct matrix G2. The genetic variance and environmental variance were 5.236 and 10.93, respectively. The heritability was 0.324. The weighted values of G1 and G2 were 0.436 and 0.564, respectively. Additionally, the genetic variance and environmental variance were 5.301 and 10.800, respectively. The heritability was 0.329, and the prediction accuracy was 0.164, representing an improvement accuracy by 6.49%. The most significant SNPs (top 20%) were selected to construct a kinship matrix G1. The genetic variance and environmental variance obtained were 4.786 and 11.319, respectively, resulting in a heritability of 0.297. The remaining 80% of SNPs were used to construct matrix G2. The genetic variance and environmental variance were 5.129 and 11.033, respectively. The heritability was 0.317. The weighted values of G1 and G2 were 0.483 and 0.517, respectively. The G3 matrix was constructed with genetic variance and environmental variance of 5.493 and 10.626, respectively. The heritability was 0.341, and the prediction accuracy was 0.166, representing a 7.79% improvement in accuracy.

The GWAS results from trial group 1 were utilized as prior information to construct a G matrix. This matrix was subsequently employed to validate trial group 2 and assess the accuracy of genome prediction. The most significant SNPs (top 5%) were selected to construct matrix G1. The genetic variance and environmental variance were 4.463 and 11.960, respectively, resulting in a heritability of 0.272. The remaining 95% SNPs were used to create matrix G2, which yielded a genetic variance of 5.968, an environmental variance of 10.542, and a heritability of 0.361. The weights of G1 and G2 were 0.428 and 0.572, respectively, which led to the construction of matrix G3. The genetic variance and environmental variance for G3 were 6.296 and 10.174, respectively, resulting in a heritability of 0.382. The prediction accuracy was 0.201, indicating a 5.79% improvement in accuracy. Furthermore, the most significant SNPs (top 10%) were selected to construct the matrix G1, resulting in a genetic variance of 4.688 and an environmental variance of 11.740. The heritability was estimated to be 0.285. The remaining 90% of SNPs were used to construct matrix G2, which had a genetic variance of 6.012, an environmental variance of 10.503, and a heritability of 0.364. The weights of G1 and G2 were 0.438 and 0.562, respectively. The genetic variance and environmental variance for matrix G3 were 6.057 and 10.340, respectively, resulting in a heritability of 0.368. The prediction accuracy was 0.188, representing a decrease in accuracy of 1.05%.

The most significant SNPs (top 15%) were selected to construct matrix G1. The resulting genetic variance and environmental variance were 4.969 and 11.475, respectively, with a heritability of 0.302. The remaining 85% of SNPs were used to construct matrix G2, which had a genetic variance of 6.027, an environmental variance of 10.489, and a heritability of 0.365. The weights of G1 and G2 were 0.452 and 0.548, respectively. The genetic variance and environmental variance for G3 were 6.060 and 10.396, respectively, resulting in a heritability of 0.368. The prediction accuracy was 0.184, resulting in a decrease in accuracy of 3.15%. Similarly, the most significant SNPs (top 20%) were selected to construct matrix G1, resulting in a genetic variance of 5.288 and an environmental variance of 11.194. The heritability was estimated to be 0.321. The remaining 80% of SNPs were used to construct matrix G2, which had a genetic variance of 5.986, an environmental variance of 10.523, and a heritability of 0.363. The weights of G1 and G2 were 0.469 and 0.531, respectively. The G3 matrix was constructed with a genetic variance of 6.139 and an environmental variance of 10.330. The heritability was calculated to be 0.373. The prediction accuracy was 0.185, resulting in a decrease in accuracy by 2.63%.

## 4. Discussion

### 4.1. Animals and Data

Alpine Merino Sheep is a new breed of Chinese sheep with exceptionally high-quality wool and great meat quality. These sheep thrive in cold, arid alpine areas throughout the year. Improving the breeding of Alpine Merino Sheep can have a positive impact on the local economy. Therefore, this study used the 14-month-old body weight trait of 1310 Alpine Merino ewes from the core breeding station of Huangcheng Sheep Breeding Extension Station in Gansu Province as the target trait for GWAS and GS analysis. The aim was to identify candidate genes associated with the target trait and determine the most suitable genomic prediction model for the trait. These phenotypic data were clean, well-documented, and free from outliers for research analysis. The genotype data used in this study was the Fine Wool Sheep 50K Panel data of the experimental animal for GWAS analysis. Following model correction, the Q-Q plot of GWAS results showed a good fit. The GBLUP model used in the GS analysis demonstrated a prediction accuracy ranging from 0.154 to 0.190. This was not significantly different from the results of genomic prediction accuracy studies on the carcass weight of the pig [26], the carcass traits of sheep [27], and the one-year weight of Hanwoo beef cattle [28]. The findings of this study were reliable.

### 4.2. GWAS and Candidate Genes

Body weight traits are typical quantitative traits regulated by micro-efficacious polygenes, and GWAS is a powerful tool for identifying the genes underlying such traits. Many researchers have previously conducted GWAS analyses on livestock with a particular emphasis on body weight traits. For example, *CAPN6*, which is associated with the birth weight of Hu Sheep, was identified using GWAS analysis [29]. Additionally, 84 genes (including *BMPR1B*, *HSD17B3*, and *TMEM63C*) related to sheep production traits were identified [30]. Furthermore, important candidate genes related to body size [31], body weight [32], and muscle growth and development [33] in goats were identified, playing a significant role in promoting genetic improvement. The GWAS analysis results of this study also provided annotations for some genes previously reported in other papers, such as *ANKRD44*, *MACF1*, *SYN3*, *DCAF16*, *FAM184B*, *NCAPG*, *LCORL*, and *FUK*.

The *FAM184B* gene is located on sheep chromosome 6, and studies have demonstrated that it is not only associated with sheep meat quality, weight, and body shape traits [34] but also with fat metabolism, skeletal system development, and meat quality traits [35]. Additionally, the *FAM184B* gene has been identified as a gene strongly associated with body size and weight traits in livestock and poultry, including donkeys [36], chickens [37,38], and Red Angus cattle [39]. The *NCAPG* gene is located on sheep chromosome 6. Posbergh determined that this gene was associated with sheep size [40], and Al-Mamun reported that it was not only related to sheep weight but also related to cattle growth, carcass size, and human height [41,42]. Liu [43] demonstrated that the *NCAPG* gene was abundant in the muscles of Qinchuan cattle. The analysis also identified that this gene plays a role in muscle growth and development. The *NCAPG–LCORL* loci were found to be significantly associated with feed intake and weight gain phenotypes in beef cattle [44]. Additionally, in related studies on other livestock species such as horses [45,46], deer [47], and Korean cattle [48], the association of the NCAPG gene with body weight and height has also been validated. It has been discovered that multiple quantitative trait loci found on chromosome 6 of sheep impact the fat area, fat density, fat weight, and muscle density [49]. The LCORL gene is associated with weight traits and embryo development in Hu sheep [8]. It is also related to height [41] in humans and cattle as well as growth and feed intake in sheep [15]. NCAPG/LCORL has been identified as being associated with body size in Chinese Merino sheep [50]. The *MACF1* gene is essential for regulating the actin and microtubule cytoskeleton network. The knockout function of mouse models revealed that *MACF1* is associated with skin integrity and bone formation [51]. This gene was identified as a candidate gene related to spine curvature in Chinese indigenous pigs [52]. The *SYN3* gene has been implicated in the body length asymmetry of goats and in the extracellular matrix of neuroregulatory processes [53]. The *ANKRD44* gene has been found to have a significant association with birth weight in pigs. Additionally, it has been validated to be significantly associated with bone mineral density and whole-body lean mass [54,55]. This gene has also been found to be significantly associated with low birth weight and postprandial triglyceride concentration in pigs, which can have an impact on early-life metabolism [54]. The *DCAF16* gene is significantly associated with the average daily gain of cattle, and the *DCAF16-NCAPG* region is a susceptible site for this trait [56]. The studies have shown that the growth traits and fertility of dairy cows were strongly correlated with the expression of the *FUK* gene [57]. These results indicate that the genes *FAM184B*, *NCAPG*, *MACF1*, *SYN3*, *ANKRD44*, *DCAF16*, *FUK*, and *LCORL* may be associated with body weight, body size, and muscle development in Alpine Merino Sheep. These genes and can be considered as potential candidates for target traits.

In addition to *FAM184B*, *NCAPG*, *MACF1*, *SYN3*, *ANKRD44*, *DCAF16*, *FUK*, and *LCORL* genes, which have been reported as relevant genes in the body weight and body shape of sheep, cows and other domestic animals in many articles, the other genes reported in this study, *SPRR4*, *ACTR2*, *SPRED2*, *TRNAC*, *TAB2*, *SF3B3*, *COG4*, *ST3GAL2*, *DDX19A*, *AARS*, *BDNF*, *FYB*, *CCND1,* and *DLGAP2,* are considered to be equally important. These genes are potential genes for sheep growth and body weight. Specifically, the *DDX19A*, *TAB2*, *ST3GAL2,* and *CCND1* genes are of particular interest. *DDX19A* is highly expressed in several mammalian tissues, including the brain smooth muscle, skeletal muscle, placenta, and fetal heart, with the highest expression observed in skeletal muscle [58]. This indicates that DDX19A plays an important role in regulating skeletal muscle growth. The studies have found that the mutation of the *TAB2* gene is related to human developmental delay [59]. The ST3GAL2 gene is associated with late-onset obesity in mice [60], and the CCND1 gene regulates muscle growth in fish by controlling the expression of related genes [61]. When miR-21-5p is highly expressed in the skeletal muscle of chickens, it is accompanied by an increase in CCND1 expression [62]. This indicates that CCND1 plays a regulatory role in the growth of chicken skeletal muscle.

### 4.3. Comparative Analysis of the Accuracy of Different Models of Genome Selection

In this study, the prediction accuracy of the GBLUP model for the two groups ranged from 0.154 to 0.190. However, when the G matrix was constructed by combining the results of GWAS as prior information, the prediction accuracy of genomic selection ranged from 0.158 to 0.201. We used the GWAS results from group 1 to predict GS in group 2 and vice versa, which helped prevent overfitting. The construction of the G matrix with GWAS results as prior information in GS is different from other genome selection prediction models. This is because GWAS analysis has already identified SNP loci that are associated with target traits, and these loci will have higher heritability when constructing the G matrix. The higher the heritability, the greater the prediction accuracy. However the prediction accuracy does not continue to increase with higher heritability. There is a limit to the increase, which is consistent with the hypothesis and findings of this study. However, in this study, when the GWAS results of group 2 were used as prior information to predict the genome prediction accuracy of group 1, there were varying degrees of improvement, specifically for the top 5%, top 10%, top 15%, and top 20%, respectively. On the other hand, when the GWAS results of group 1 were used as prior information to predict group 2, only the accuracy of the top 5% SNPs improved, while the other parts showed a decline. It could be due to the following reasons: (1) the site effect screened by group 2 is larger than that of group 1, and (2) the sample size of this study is relatively small. Despite the interference of other factors, the accuracy of genome prediction is improved by constructing a G matrix based on GWAS prior information compared to traditional breeding methods. This improvement has a positive impact on enhancing the accuracy of genome prediction. It is unreasonable to assume that all SNPs in the GBLUP model have the same effect and follow the same distribution. Some researchers have integrated the GS model with prior information and made various advancements, including the use of genomic feature BLUP [63] and Bayes RC [64]. Zhang [65] proposed a method called TABLUP, which optimizes the genome relationship matrix and constructs a trait-specific relationship matrix. TABLUP assigns different weights to all markers based on the best linear unbiased prediction methods, namely indirect Bayes and ridge regression. This method shows a significant improvement over GBLUP in terms of prediction accuracy. However, the Bayes model has a long calculation time, high complexity, and large memory consumption, making it unsuitable for big data analysis. Therefore, based on the results of GWAS analysis, the SNP sites were sorted by *p*-value. The SNPs with a significant impact on their phenotype were compared to traditional breeding methods. This improved the G matrix. SNPs associated with the target traits have been partially screened out, and as a result, the study finding are consistent with expectations. Compared to the GBLUP model, this method can significantly enhance prediction accuracy.

## 5. Conclusions

In this study, Alpine Merino Sheep ewes were used as the research group, and the target trait was initially analyzed using a GWAS at 14 months of age. Subsequently, the accuracy of estimating genomic breeding values, when combined with prior information from GWASs, was compared to the GBLUP model for estimating the breeding values of target traits. The GWAS analysis using the FarmCPU model identified eight candidate genes for the target trait: *FAM184B*, *NCAPG*, *MACF1*, *ANKRD44*, *DCAF16*, *FUK*, *LCORL*, and *SYN3*. Additionally, SNPs representing the top 5% to top 20% based on the *p*-value in the GWAS analysis were selected as prior information. The G matrix was then constructed using the GBLUP model in GS analysis. The results of genetic parameter estimation and prediction accuracy analysis for the WT trait demonstrated that the accuracy of breeding values improved by 7.79% when the top 20% SNPs were used as prior information. Consequently, based on the results of the GWAS analysis used as prior information, the accuracy of genomic prediction achieved by constructing the G matrix in GS analysis was higher than that of the GBLUP model. This suggests that this method is suitable for selecting weight traits at 14 months of age in this population.

## Figures and Tables

**Figure 1 animals-13-03516-f001:**
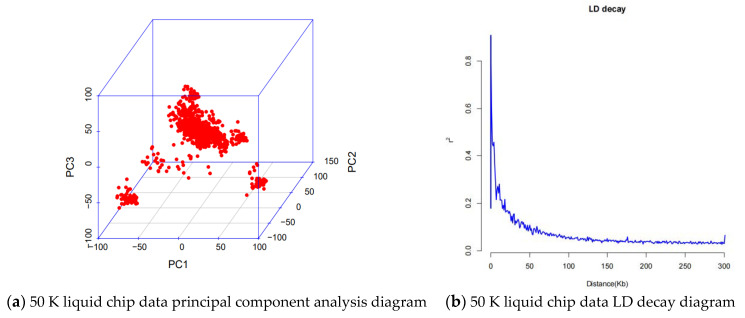
(**a**) Figure indicates that there may be stratification in the population; (**b**) Figure indicates that LD gradually flattens out when the linkage disequilibrium is equal to 100 kb.

**Figure 2 animals-13-03516-f002:**
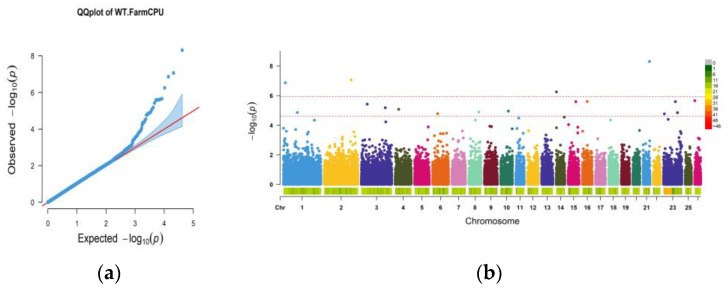
(**a**) represents the Manhattan plot in the genome-wide association studies (GWAS) results, and the starting point in the figure is around 10^−3^; (**b**) represents the QQ plot in the GWAS result, which sets the two threshold lines (red lines) of 0.05 and 1, respectively.

**Table 1 animals-13-03516-t001:** Descriptive statistics of WT.

Item	Number	Mean	Sd	Median	Trimmed	Mad	Min	Max	Se
All	1310	44.79	4.35	44.4	44.66	4.15	32.6	58.6	0.12
Group 1	655	44.72	4.38	44.4	44.63	4.45	33.4	57.8	0.17
Group 2	655	44.86	4.31	44.4	44.69	4.15	32.6	58.6	0.17

**Table 2 animals-13-03516-t002:** Gene list of single nucleotide polymorphisms (SNPs) associated with live weight in the 14th month.

Trait	Chr	SNP Name	Position	*p*-Value	Gene Name	Gene Position
WT	1	1_13705982	13,705,982	1.38 × 10^−7^	*MACF1*	13,541,086–13,881,011
	1	1_101812709	101,812,709	1.39 × 10^−5^	*SPRR4*	101,734,532–101,736,907
	2	2_198504883	198,504,883	8.74 × 10^−8^	*ANKRD44*	198,238,519–198,479,012
	3	3_42773188	42,773,188	3.76 × 10^−6^	*ACTR2*	42,872,405–42,911,004
	3	3_42773188	42,773,188	3.76 × 10^−6^	*SPRED2*	42,708,519–42,827,598
	3	3_176412255	176,412,255	6.67 × 10^−6^	*TRNAC*	176,470,024–176,470,095
	3	3_176412255	176,412,255	6.67 × 10^−6^	*SYN3*	175,963,647–176,440,791
	6	6_37211546	37,211,546	1.65 × 10^−5^	*DCAF16*	37,199,219–37,211,835
	6	6_37211546	37,211,546	1.65 × 10^−5^	*FAM184B*	37,061,089–37,179,847
	6	6_37211546	37,211,546	1.65 × 10^−5^	*NCAPG*	37,179,188–37,257,373
	6	6_37211546	37,211,546	1.65 × 10^−5^	*LCORL*	37,274,935–37,426,290
	8	8_73354847	73,354,847	1.29 × 10^−5^	*TAB2*	73,362,589–73,448,254
	14	14_1078888	1,078,888	5.70 × 10^−7^	*SF3B3*	957,834–996,615
	14	14_1078888	1,078,888	5.70 × 10^−7^	*COG4*	996,835–1,022,199
	14	14_1078888	1,078,888	5.70 × 10^−7^	*FUK*	1,022,451–1,036,935
	14	14_1078888	1,078,888	5.70 × 10^−7^	*ST3GAL2*	1,052,461–1,101,237
	14	14_1078888	1,078,888	5.70 × 10^−7^	*DDX19A*	1,110,280–1,128,800
	14	14_1078888	1,078,888	5.70 × 10^−7^	*AARS*	1,165,135–1,185,981
	15	15_56544782	56,544,782	2.56 × 10^−6^	*BDNF*	56,424,495–56,488,905
	16	16_35207312	35,207,312	2.49 × 10^−6^	*FYB*	35,105,425–35,282,410
	21	21_45980537	45,980,537	2.96 × 10^−9^	*CCND1*	46,064,385–46,074,757
	26	26_499128	499,128	2.22 × 10^−6^	*DLGAP2*	279,898–905,775

**Table 3 animals-13-03516-t003:** Genetic parameters of group 1.

Prior Marker Information	Matrix	Genetic Variance	Environmental Variance	Heritability	Weight	Prediction Accuracy	Promotion
-	G	5.427	10.751	0.335	-	0.154 (0.03)	-
Top 5%	G1	2.702	13.308	0.169	0.333	-	-
	G2	5.418	10.764	0.335	0.667	-	-
	G3	4.840	11.229	0.301	-	0.158 (0.03)	+2.59%
Top 10%	G1	4.035	12.013	0.251	0.434	-	-
	G2	5.256	10.915	0.325	0.566	-	-
	G3	5.331	10.767	0.331	-	0.165 (0.02)	+7.14%
Top 15%	G1	4.052	11.992	0.253	0.436	-	-
	G2	5.236	10.933	0.324	0.564	-	-
	G3	5.301	10.800	0.329	-	0.164 (0.03)	+6.49%
Top 20%	G1	4.786	11.319	0.297	0.483	-	-
	G2	5.129	11.033	0.317	0.517	-	-
	G3	5.493	10.626	0.341	-	0.166 (0.03)	+7.79%

**Table 4 animals-13-03516-t004:** Genetic parameters of group 2.

Prior Marker Information	Matrix	Genetic Variance	Environmental Variance	Heritability	Weight	Prediction Accuracy	Promotion
-	G	6.176	10.351	0.374	-	0.190 (0.02)	-
Top 5%	G1	4.463	11.960	0.272	0.428	-	-
	G2	5.968	10.542	0.361	0.572	-	-
	G3	6.296	10.174	0.382	-	0.201 (0.02)	+5.79%
Top 10%	G1	4.688	11.740	0.285	0.438	-	-
	G2	6.012	10.503	0.364	0.562	-	-
	G3	6.057	10.340	0.368	-	0.188 (0.02)	−1.05%
Top 15%	G1	4.969	11.475	0.302	0.452	-	-
	G2	6.027	10.489	0.365	0.548	-	-
	G3	6.060	10.396	0.368	-	0.184 (0.02)	−3.15%
Top 20%	G1	5.288	11.194	0.321	0.469	-	-
	G2	5.986	10.523	0.363	0.531	-	-
	G3	6.139	10.330	0.373	-	0.185 (0.02)	−2.63%

## Data Availability

Data will be available upon request from the corresponding author.

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
