# Peer review of "Genomic Selection for Live Weight in the 14th Month in Alpine Merino Sheep Combining GWAS Information"

_animals, 2023, doi:10.3390/ani13223516_

Round 1

Reviewer 1 Report (Previous Reviewer 2)

Comments and Suggestions for Authors

 I would like to thank the authors for their efforts to improve the manuscript, but I still have some issues that could be improved. The main ones are listed below, other minor ones have been flagged up in the attached pdf.

Line 79 – 81

“Body weight traits are closely related to meat quality traits and economic benefits, and accelerating the breeding of body weight traits can not only improve meat quality, but also increase local economic income.”

The authors introduced this statement that I'm not entirely comfortable with. Depending on consumer preferences, heavier/fatter animals may have fatter carcasses and meat with more intense sheep flavours and tougher tendencies, which may or may not be more appreciated/valued. So, this is a statement that deserves reflection and a thorough review. I hadn't felt comfortable with it in the summary, but the way the problem has been put here is not scientifically valid.

Line 97-98

“Accelerating the selective breeding of Alpine Merino Sheep for weight traits and shorten the gap with Australian Merino Sheep.”

Please review the sentence. Something is missing.

Line 111

each sheep was kept better than the automatic weight scale for weighing”

I don’t understand???

Line 253-259

and the PCA results were assessed using the BLINK model in the GAPIT software (version 3). The PCA diagram is exhibited in Figure 3 indicating stratification in the experimental population. Therefore, using PLINK 1.9 software (version number: v1.9b4) to calculate PCA and further EIGENSTRAT software to calculate PCA significance, respectively, the calculation results demonstrated that the first nine PCAs reached the P<0.05 significance level, and finally, the first nine principal components were added to the model as covariates in subsequent analyses.”

This should have been referred in the material and methods session.

Line 411-413
“Additionally, in other livestock and poultry studies, researchers have identified NCAPG genes, and horse[40,41], deer[42], and Korean cattle[43]”

Please review

Linr 482-485

“The Farm-CPU model used by GWAS analysis discovered ten SNP significance sites and 15 genes associated with the target trait and identified five candidate genes for the target trait, namely the FAM184B, NCAPG, MACF1, ANKRD44, DCAF16, FUK, DDX19A, and SYN3 genes.”

This is very confusing: 15 genes, 10 SNPs, 5 candidate genes, but the author cited by name 8 genes... Please clarify!

Line 559-590

These papers are not numbered. Are they cited? If so, they should be numbered, if not delete the references.

Comments on the Quality of English Language

I still detected some minor problems with sentence construction and English, which I've noted in the pdf.

Author Response

Reviewer 2 Report (Previous Reviewer 1)

Comments and Suggestions for Authors

the manuscript has been successfully revised and now it is suitable for pubblication 

Author Response

Reviewer 3 Report (New Reviewer)

Comments and Suggestions for Authors

Dear authors,

I will recommend your manuscript for publication, but I has some questions for edition.

L2 – Change “Weight” to “Live Weight in 14-th Months”, remove “Prior Marker”

L108 – not clear aim of group dividing

L111 – show accuracy of automatic weight scale

L117 – change “sampling” to “puncturing”

L120 – change “for later genotyping” to “for later DNA extraction”

L121 – What is reference genome using for this DNA-array?

L123 - < 10 E-6 is wrong format

L130 – How many samples passed quality control?

L155 – not clear – what is threshold 1? And how are you taken threshold 2.64 × 10−7 with Bonf correction for 41987 SNPs in set?

L161 – why you are select “50 kb regions” for genes searching?

L186 – Don’t use “W” in formulas, because “W” is more associated with “Weight” abbreviation

L203 – What is Table 1?

L216 – Who was recommend this method for the accuracy calculating?

L234 – This is part of Methods

L239-240 – This is repeat.

L248 – This Figure is not your result. It is datasheet for DNA-array.

L253 – PCA not presented in Methods

L257 – What you are mean in “first nine PCAs reached the P<0.05 significance level”? PCA is just reduce data dimension, but not statistical method for significance detecting.

L272 – I can’t see SNP in 2, 8 and 14 Chromosomes in Manhattan plot.

L281 – Not clear criterions of candidate genes selection

L282 – What is the heat scale in Manhattan plot?  Not clear threshold calculation in plot.

L285 – “represents the QQ plot in the GWAS result, which sets the two threshold lines (red lines) 285 of 0.05 and 1, respectively” – I can’t see lines in QQ-plot.

L287 – I recommend add in table distance to gene or localization in gene

L298 and table – not clear, what is matrix G, it was not presented in Methods. Also, not clear who recommended this method with select % of SNPs for accuracy analysis.

L436 – Too long and not clear sentence

L556 – Wrong reference format

Regards,

Author Response

Reviewer 4 Report (New Reviewer)

Comments and Suggestions for Authors

Now I have concerned about the choice of statistical model. For example, how did the authors consider the macro-environment effects, including year, month, farm, and so on? Also, how did the authors determine the values for variance components to calculated γ in G3? Results obtained using only G1 or G2 could affect the LD between SNPs considered in the respective G matrix and those not considered. Furthermore, there would be over-fitting in the results obtained by using G1. This could be related the fact that the sum of the estimated values of genetic variances by using G1 and G2 was greater than those estimated using G and even G3. All of them might bring very low prediction accuracy (~0.2) relative to moderate heritability (~0.35).

Comments on the Quality of English Language

Moderate English check would be recommended.

Round 2

Reviewer 3 Report (New Reviewer)

Comments and Suggestions for Authors

Dear authors,

I will recommend to publish your manuscript.

Best regards,

Author Response

Reviewer 4 Report (New Reviewer)

Comments and Suggestions for Authors

Unfortunately, my opinions/comments were not well received by the authors.

Author Response

This manuscript is a resubmission of an earlier submission. The following is a list of the peer review reports and author responses from that submission.

Round 1

Reviewer 1 Report

Comments and Suggestions for Authors

In the study entitled “Genomic Selection for Weight in Alpine Merino Sheep Combining GWAS Prior Marker Information” Authors used genome-wide association studies (GWAS) to screen out the variation information related to the 14-month-old body weight trait of Alpine Merino Sheep, identify candidate genes associated with this trait, and use the results of GWAS as prior information to conduct genome selection research on the trait to screen out the genomic selection breeding model most suitable for the target trait.

Overall, the study is well-written; the study protocol was meticulously described and the obtained findings were well-defined and discussed. I believe that the study is interesting in its field, that it well fits into the purpose of the journal, indeed, it may capture the attention of its readers.

Specific Comments

The title well reflects the major findings of the study.

I suggest to avoid the use of personal form (i.e. our, we etc.) throughout the text.

The abstract adequately summarize methodology, results, and significance of the study. However, Authors should better emphasize the significance of the study in a conclusive sentence in the abstract section.

The introduction section is well written and it falls within the topic of the study.

The section of Materials and Methods is clear for the reader and well describes the methods applied in the study.  However, Authors should check this section and correct many punctuation errors.

Results section as well as Discussion section is clear and well written. The findings obtained in the study were well discussed and justified with appropriate references. However, in order to make more harmonious the discussion, I suggest, to split the section Results and Discussion into two separated sections, one for results description and the other one for the discussion of findings.

The conclusion section is well written, Authors well summarize the results and the significance of the study.

The Figures are nice and well represent the results of the study.

Author Response

Response to Reviewer 1Comments

Thank you for your comments and suggestions concerning our manuscript. The comments and suggestions are all valuable and very helpful for revising and improving our paper, as well as the important guiding significance to our research. We have studied the comments carefully and have made corrections which we hope meet with approval. The following is a response to each of your comments.

Point 1: Suggest avoiding the use of personal form (i.e. our, we, etc.) throughout the text, Check the Materials and Methods section for many incorrect punctuation marks and make corrections.

Response 1: From the beginning of receiving your comments, we will re-check and correct the English writing and punctuation of the article according to your comments, so that the article looks more smooth and harmonious.

Point 2: In order to make more harmonious the discussion, I suggest, to split the section Results and Discussion into two separate sections.

Response 2: We wrote the results and discussion separately to make the results more summative, strengthen the discussion of genome-wide association analysis and genome selection, and add some references related to this study to the discussion section.

Reviewer 2 Report

Comments and Suggestions for Authors

In this work, the authors used GWAS to identify candidate genes associated with14-month-old body weight trait of Alpine Merino Sheep.

Across the introduction, the authors didn’t refer to any study using this approach in sheep or in other livestock specie. Thus, I suggest a major review of the introduction section.

Throughout the text, the authors cited several studies regarding chickens (refª 10, 23, 24) and failed to refer to more relevant studies e.g. in:

Sheep:

Zhang, L., Liu, J., Zhao, F., Ren, H., Xu, L., Lu, J., ... & Du, L. (2013). Genome-wide association studies for growth and meat production traits in sheep. PloS one, 8(6), e66569.

Kominakis, A., Hager-Theodorides, A. L., Zoidis, E., Saridaki, A., Antonakos, G., & Tsiamis, G. (2017). Combined GWAS and ‘guilt by association’-based prioritization analysis identifies functional candidate genes for body size in sheep. Genetics Selection Evolution, 49(1), 1-16.

Yurchenko, A. A., Deniskova, T. E., Yudin, N. S., Dotsev, A. V., Khamiruev, T. N., Selionova, M. I., ... & Larkin, D. M. (2019). High-density genotyping reveals signatures of selection related to acclimation and economically important traits in 15 local sheep breeds from Russia. BMC genomics, 20, 1-19.

Ghasemi, M., Zamani, P., Vatankhah, M., & Abdoli, R. (2019). Genome-wide association study of birth weight in sheep. Animal, 1–7. doi:10.1017/s1751731118003610

Matika O. et al. Genome-wide association reveals QTL for growth, bone and in vivo carcass traits as assessed by computed tomography in Scottish Blackface lambs. Genetics, Selection, Evolution: GSE 48, 11, doi: 10.1186/s12711-016-0191-3IF: 5

Cao, Y., Song, X., Shan, H., Jiang, J., Xiong, P., Wu, J., ... & Jiang, Y. (2020). Genome-wide association study of body weights in Hu sheep and population verification of related single-nucleotide polymorphisms. Frontiers in genetics, 11, 588.

Li, X., Yang, J. I., Shen, M., Xie, X. L., Liu, G. J., Xu, Y. X., ... & Li, M. H. (2020). Whole-genome resequencing of wild and domestic sheep identifies genes associated with morphological and agronomic traits. Nature communications, 11(1), 2815.

Tuersuntuoheti, M., Zhang, J., Zhou, W., Zhang, C. L., Liu, C., Chang, Q., & Liu, S. (2023). Exploring the growth trait molecular markers in two sheep breeds based on Genome-wide association analysis. Plos one, 18(3), e0283383.

Mohammadi, H., Raffat, A., Moradi, H., Shoja, J., & Moradi, M. H. (2018). Estimation of linkage disequilibrium and whole-genome scan for detection of loci under selection associated with body weight in Zandi sheep breed. Agricultural Biotechnology Journal, 9(4), 151-172.

Krivoruchko, A., Surov, A., Kanibolotskaya, A., Sheludko, P., Likhovid, N., Yatsyk, O., & Likhovid, A. (2023). A genome-wide search of meat productivity candidate genes in Russian Meat Merino breed. Animal Gene, 27, 200146.

Ma, Q., Liu, X., Pan, J., Ma, L., Ma, Y., He, X., ... & Jiang, L. (2017). Genome-wide detection of copy number variation in Chinese indigenous sheep using an ovine high-density 600 K SNP array. Scientific reports, 7(1), 912.

Goats:

Moaeen-ud-Din, M., Danish Muner, R., & Khan, M. S. (2022). Genome wide association study identifies novel candidate genes for growth and body conformation traits in goats. Scientific Reports, 12(1), 1-12.

Selionova, M., Aibazov, M., Mamontova, T., Malorodov, V., Sermyagin, A., Zinovyeva, N., & Easa, A. A. (2022). Genome-wide association study of live body weight and body conformation traits in young Karachai goats. Small Ruminant Research, 216, 106836.

Zhang, L., Wang, F., Gao, G., Yan, X., Liu, H., Liu, Z., ... & Su, R. (2021). Genome-Wide Association Study of Body Weight Traits in Inner Mongolia Cashmere Goats. Frontiers in Veterinary Science, 1246.

Easa, A. A., Selionova, M., Aibazov, M., Mamontova, T., Sermyagin, A., Belous, A., ... & Zinovieva, N. (2022). Identification of Genomic Regions and Candidate Genes Associated with Body Weight and Body Conformation Traits in Karachai Goats. Genes, 13(10), 1773.

Cattle:

Xia, J., Fan, H., Chang, T., Xu, L., Zhang, W., Song, Y., ... & Gao, H. (2017). Searching for new loci and candidate genes for economically important traits through gene-based association analysis of Simmental cattle. Sci. Rep. 7, 42048; doi: 10.1038/srep42048

Lindholm-Perry A. K. et al. (2011). Association, effects and validation of polymorphisms within the NCAPG - LCORL locus located on BTA6 with feed intake, gain, meat and carcass traits in beef cattle. BMC genetics 12, 103, doi: 10.1186/1471-2156-12-103

Setoguchi K. et al. The SNP c.1326T>G in the non-SMC condensin I complex, subunit G (NCAPG) gene encoding a p.Ile442Met variant is associated with an increase in body frame size at puberty in cattle. Animal genetics 42, 650–655, doi: 10.1111/j.1365-2052.2011.02196.xIF: 2.884 Q1 (2011).

2.1. Phenotype Data

Why did you phenotype only the females?

2.2. Genotype Data

Did the sheep 50K SNP chip (MolBreeding Biotech Ltd., China) be designed considering Chinese sheep breed-specific SNPs? Its development has been published in the paper intituled “Yingwei Guo and others, Design and characterization of a high-resolution multiple-SNP capture array by target sequencing for sheep, Journal of Animal Science, Volume 101, 2023, skac383, https://doi.org/10.1093/jas/skac383”?

Do you have information about how many SNPS are simultaneously present in the sheep 50K SNP chip (MolBreeding Biotech Ltd., China) and in the, e.g., Illumina Ovine SNP 600 BeadChip (Illumina Inc., CA, USA) or other chip? How does it compare to the other chips? Do you think the same results could be obtained using another ovine chip?

3. Results and Discussion

Figure 1 – could be presented as supplemental material.

Values presented in Table 2 could be presented in the body of the text.

Present the results reported in Lines 228 to 233 as a table.

Line 267

“body size and weight traits of livestock and poultry such as donkeys[22], chickens[23,24], and Red Angus cattle[25].”

What about other sheep breeds? And goat breeds? - The same question for the other genes.

The author mentioned chickens several times. But what about pigs?

The author didn’t discuss the possible importance to WT of several genes presented in Table 3, e.g.:

SYN3 gene

SYN3 has been reported as a candidate gene associated with body measurement traits in Chinese Wagyu beef cattle. (An, B., Xia, J., Chang, T., Wang, X., Xu, L., Zhang, L., ... & Gao, H. (2019). Genome‐wide association study reveals candidate genes associated with body measurement traits in Chinese Wagyu beef cattle. Animal genetics, 50(4), 386-390.)

MACF1 gene

MACF1 gene was identified as a candidate gene related to spine curvature in Chinese indigenous pigs. (Jiayuan M, Yujie L, Kuirong C, Siran Z, Wenjing Q, Lingli F, Xiaoxiao L, Liang L, Ganqiu L, Jing L. Identifying selection signatures and runs of homozygosity for spine curvature in Chinese indigenous pigs. Anim Genet. 2022 Aug;53(4):513-517. doi: 10.1111/age.13224. Epub 2022 May 30. PMID: 35634679.)

SLC19A2 gene

Cold exposure caused genes to be upregulated namely the SLC19A2 in pigs (Zhang D, Ma S, Wang L, Ma H, Wang W, Xia J, Liu D. Min pig skeletal muscle response to cold stress. PLoS One. 2022 Sep 26;17(9):e0274184. doi: 10.1371/journal.pone.0274184).

The authors did not discuss the possible importance that adaptation to cold temperatures might have in ewe weight. They only considered FAM184B and NCAPG genes as possible candidate genes for WT.
WT is a complex trait.
In goats, it has been observed that SLC19A2 is differentially expressed colon epithelial in response to dietary supplementation of thiamine that enhances colonic integrity and modulates mucosal inflammation injury in goats challenged by lipopolysaccharide and low pH (Ma Y, Wang C, Elmhadi M, Zhang H, Liu F, Gao X, Wang H. Dietary supplementation of thiamine enhances colonic integrity and modulates mucosal inflammation injury in goats challenged by lipopolysaccharide and low pH. Br J Nutr. 2022 Jan 21:1-11. doi: 10.1017/S0007114522000174IF: 4.125 Q3 . Epub ahead of print. PMID: 35057872.). Mutations in this gene might be related whit enhanced feed efficiency/resistance to colon inflammation also in sheep, and thus with high daily body gain/WT.

3.5.2. Comparison Based on the Accuracy of GWAS Prior Information

I would like to see a table with the most significant SNPs as supplemental material.

Comments on the Quality of English Language

Author Response

Response to Reviewer 2 Comments

Thank you for your comments and suggestions concerning our manuscript. The comments and suggestions are all valuable and very helpful for revising and improving our paper, as well as the important guiding significance to our research. We have studied the comments carefully and have made corrections which we hope meet with approval. The following is a response to each of your comments.

Point 1: Across the introduction, the authors didn’t refer to any study using this approach in sheep or in other livestock specie. Thus, I suggest a major review of the introduction section.

Response 1: In the introduction part, we add some references listed by reviewers, and put the rest in the discussion section, which is closely related to this study.

Point 2: Why did you phenotype only the females? How does Genotype data compare to the other chips? Do you think the same results could be obtained using another ovine chip?

Response 2: The objects of this study are all ewes, and the economic traits of ewes are also crucial in breeding. This is to screen out the best model related to the weight traits of an ewe at 14 months of age. In the next study, we will introduce RAMS for related research. The genotype data in this study is the sheep chip data developed by our unit. Compared with other chips, each chip has its advantages and disadvantages. I believe that different chip data will produce different but similar results.

Point 3: Figure 1 – could be presented as supplemental material. Values presented in Table 2 could be presented in the body of the text. Present the results reported in Lines 228 to 233 as a table. Discuss all the genes in Table 3 to discuss more possibilities.

Response 3: We modified and replaced Figure 1, Table 2, and Table 4 according to the reviewer's opinions and suggestions, added Table 3, especially the genes in Table 4, and cited new references for MACF1, SLC19A2, and SYN3 genes to discuss more possibilities.

Reviewer 3 Report

Comments and Suggestions for Authors

Li et al. identified some candidate genes for body weight at 14 months of age in Alpine Merino sheep and indicated that the results of GWAs could improve the accuracy of genomic predictions.

The GWAS section is fine, and the authors used reliable methods and software for analyses although the sample size is not ideal.  I have a concern about confounding effects in the GWAs model as any confounding effects such as year of measure body weight, the initial body weight of sheep, family relationship, etc. The authors also should provide information in the manuscript or should apply some corrections for these effects, if existing.

My main concern is the genomic prediction results. The increase in the accuracy by combining the GWAS prior information is too good to be true. Are any supporting references for a formula for computing the accuracy? It seems the sample size is not enough or some biases in the analyses. I might be wrong, but I suggest the authors perform some additional tests to verify the results.

1.       Increases to 10-fold cross-validations.

2.       Add the standard errors of mean and accuracy ranges of 10-fold cross-validations.

3.       Add the prediction biases for genomic prediction methods,

4.       Make the data publicly available so other researchers could verify the results.

The authors should also extend the discussion of why the accuracy of genomic predictions significantly improved.

The authors do not need table 2 and 4, if it is only one row. Now need for the first column in table 3, as the authors analyzed only one trait. Table 5 is not making sense, why did the authors re-estimate the heritability of the trait?

Comments on the Quality of English Language

Quality of writing is fine. 

Author Response

Response to Reviewer 3 Comments

Thank you for your comments and suggestions concerning our manuscript. The comments and suggestions are all valuable and very helpful for revising and improving our paper, as well as the important guiding significance to our research. We have studied the comments carefully and have made corrections which we hope meet with approval. The following is a response to each of your comments.

Point 1: Increases to 10-fold cross-validations. Add the standard errors of mean and accuracy ranges of 10-fold cross-validations. Add the prediction biases for genomic prediction methods.

Response 1: According to the comments and suggestions of reviewers, we increased the 5-fold cross-validation of all genome selection prediction models to 10-fold cross-validation, and increased the standard error value and prediction deviation value of cross-validation, and the results were presented in the revised draft. All the data studied in this study are publicly available.

Point 2: The authors should also extend the discussion of why the accuracy of genomic predictions significantly improved. The authors do not need tables 2 and 4 if it is only one row.

Response 2: Based on the comments of the reviewers, we re-discussed the GWAS results in the discussion section as priori information to significantly improve the accuracy of the forecast, all of which are presented in the revised draft. In addition, for tables with only one line, we will not allow it to appear in the article and put it in the supplementary material.

Round 2

Reviewer 2 Report

Comments and Suggestions for Authors

I would like to thank the authors for their efforts in reformulating the article, which have greatly improved its quality.

However, I still have an issue with figure 5 which the authors have replaced with a new one where repeated information appears, making it more confusing and difficult to read. 

Comments on the Quality of English Language

There are still minor issues.

Author Response

Thank you very much for your comments and suggestions for our manuscript again. The comments and suggestions are all valuable and very helpful for revising and improving our paper, as well as the important guiding significance to our research. We have studied the comments carefully and have made corrections which we hope meet with approval. The following is a response to each of your comments.

Point 1: However, I still have an issue with Figure 5 which the authors have replaced with a new one where repeated information appears, making it more confusing and difficult to read.

Response 1: I have replaced Figure 5 again to make it more clearly visible. In addition, I have included the accuracy and unbias of Figure 5 in the supplementary materials. Thank you again for your comments.

Point 2: The Quality of English Language is still minor issues.

Response 2: We regret there were problems with the English Language. The manuscript has been carefully revised by a native English speaker of relevant majors to improve the grammar and readability, and I hope to get your approval.

Reviewer 3 Report

Comments and Suggestions for Authors

The authors did not provide the responses for how the accuracy and prediction biasness were calculated. The formula for calculation of accuracy is in correct as it does not account for changing of heritability due to changing in genomic matrix. 

Author Response

Response to Reviewer 3 Comments

Thank you very much for your comments and suggestions for our manuscript again. The comments and suggestions are all valuable and very helpful for revising and improving our paper, as well as the important guiding significance to our research. We have studied the comments carefully and have made corrections which we hope meet with approval. The following is a response to each of your comments.

Point 1: The authors did not provide the responses for how the accuracy and prediction biasness were calculated. The formula for calculation of accuracy is in correct as it does not account for changing of heritability due to changing in genomic matrix.

Response 1: First of all, thank you again for your comments and suggestions, which I believe will make the manuscript more complete. In the materials and methods of the manuscript, we have provided the formula for the calculation accuracy of this study, so it is not mentioned again in the discussion part. In this study, the accuracy of genomic predictions was obtained by calculating the Pearson Correlation coefficient between GEBV and TBV. In fact, when we combined the results of GWAS as prior information to build the matrix in GS, we first calculated the heritability of G1 (5% SNP) and G2 (95% SNP) respectively, the genomic information did not change, and assigned different weights according to the heritability to build the G3 matrix, perhaps because I did not express it clearly in the document.

Round 3

Reviewer 3 Report

Comments and Suggestions for Authors

I still have a concern about the heritability calculated for different G matrices. There are around 10 times more than the original one, indicating that The genetic variances obtained by different G matrices are overestimated. Since the authors did not give the prediction bias and other methods to check the results.  

What did the authors mean by "hereditary variance" .

The discussion did not extend why the significant increases by adjusting G matrix. 

Comments on the Quality of English Language

Need to be improved

Author Response

Response to Reviewer 3 Comments

Thank you very much for your comments and suggestions for our manuscript again. The comments and suggestions are all valuable and very helpful for revising and improving our paper, as well as the important guiding significance to our research. We have studied the comments carefully and have made corrections which we hope meet with approval. The following is a response to each of your comments.

Point 1: I still have a concern about the heritability calculated for different G matrices. There are around 10 times more than the original one, indicating that The genetic variances obtained by different G matrices are overestimated. Since the authors did not give the prediction bias and other methods to check the results.

Response 1: In fact, the construction of G matrix with GWAS results as prior information in GS is different from other genome selection prediction models, because GWAS analysis has screened out SNP loci associated with target traits, and these loci will obtain higher heritability when constructing G matrix. The higher the heritability, the higher the prediction accuracy, but the prediction accuracy does not increase with the increase of heritability, which has a limit, which is consistent with the hypothesis and results of this study. In addition, in the second half of this study, only the standard errors of accuracy and accuracy were calculated, and unbiasedness was not calculated. I will provide the standard error of accuracy in the supplementary material. I hope this reply will meet with your approval.

Point 2: What did the authors mean by "hereditary variance".

Response 2: I am very sorry for this error, probably due to a hereditary problem, but I re-examined the position and meaning of the word "hereditary variance" in the text and found that it actually meant "genetic variance".

Point 3: The discussion did not extend why the significant increases by adjusting G matrix.

Response 3: As to why accuracy has improved so significantly, as I replied in Response1, I will integrate this into the discussion again.

Point 4: The Quality of the English Language needs to be improved.

Response 4: We regret there were problems with the English Language. The manuscript has been carefully revised by Professional institutions of relevant majors to improve the grammar and readability, and I hope to get your approval.